# Chemical Compounds, Antitumor and Antimicrobial Activities of Dry Ethanol Extracts from *Koelreuteria paniculata* Laxm

**DOI:** 10.3390/plants10122715

**Published:** 2021-12-10

**Authors:** Tsvetelina Andonova, Yordan Muhovski, Hafize Fidan, Iliya Slavov, Albena Stoyanova, Ivanka Dimitrova-Dyulgerova

**Affiliations:** 1Department of Botany and Methods of Biology Teaching, Faculty of Biology, University of Plovdiv, Paisii Hilendarski, 4000 Plovdiv, Bulgaria; ts_andonova@uni-plovdiv.bg (T.A.); ivadim@uni-plovdiv.bg (I.D.-D.); 2Life Sciences Department, Walloon Agricultural Research Centre, 5030 Gembloux, Belgium; 3Department of Tourism and Culinary Management, Faculty of Economics, University of Food Technologies, 4000 Plovdiv, Bulgaria; hfidan@abv.bg; 4Department of Biology, Faculty of Pharmacy, Medical University of Varna, 9000 Varna, Bulgaria; ijelev80@abv.bg; 5Department of Tobacco, Sugar, Vegetable and Essential Oils, Faculty of Technology, University of Food Technologies, 4000 Plovdiv, Bulgaria; aastst@abv.bg

**Keywords:** *Koelreuteria paniculata*, dry ethanol extracts, GC-MS analysis, chemical compounds, antitumor and antimicrobial activities

## Abstract

*Koelreuteria paniculata* Laxm. is used in traditional medicine and has various established biological activities, however, the species is considered to be a potentially invasive alien tree species for Bulgarian flora. However, there is still much to be studied about the phytochemical and biological characteristics of the species. The present study aimed to determine the chemical composition of the ethanol extracts of aerial plant parts, by GC-MS analysis, and to thereby evaluate their in vitro antitumor and antibacterial properties. All three extracts were tested against the HT-29 and PC3 tumor cell lines using the MTT assay. Fifty-six components were identified from leaf, flower, and stem bark extracts, and over 10% were the following constituents: pyrogallol, *α*-terpinyl acetate, neryl acetate, and *α*-terpinyl isobutanoate. The oxygenated monoterpenes predominated in the extracts, followed by the oxygenated aliphatics and phenylpropanoids. Significant antiproliferative activity on the HT-29 cell line (IC_50_–21.44 µg/mL and 23.63 µg/mL, respectively) was found for the flower and leaf extracts. Antibacterial activity was established for the following bacteria strains: *Bacillus subtilis* ATCC 6633, *Bacillus cereus* NCTC 10320, *Escherichia coli* ATCC 8739, *Pseudomonas aeruginosa* ATCC 6027, and *Proteus vulgaris* ATCC 6380. The stem bark and flower extracts showed better antimicrobial potential. *K. paniculata* could be considered as a potential source of biologically active substances with antitumor and antibacterial properties.

## 1. Introduction

Nowadays, more and more authors are studying the chemical composition of various plant extracts in order to find new sources, of plant origin, to combat resistant human pathogens and cancers. Emerging allergies and the many side effects of synthetic drugs are grounds to look for their natural alternatives that are sufficiently effective and, at the same time, less harmful to human health [1,2,3,4,5,6,7,8]. Antibiotic-resistant bacteria are a significant modern problem. The key to solving it can be individual plant compounds, essential oils, or extracts containing some of the most active phytochemicals with antibacterial activity, such as polyphenols and terpenes [8]. Many natural plant compounds also exhibit potent anti-cancer activity. The therapeutic value of herbal sources in the fight against cancer has increased in recent years worldwide, as evidenced by the use of certain chemotherapeutic drugs isolated from medicinal plants [6].

The present study was focused on the evaluation of the biologically active components of possible natural sources, such as *Koelreuteria paniculata* Laxm. (belonging to the family of Sapindaceae), known as the golden rain tree. The species is native to China, but it is widely used for ornamental purposes in Europe, including Bulgaria. It is now considered to be an invasive alien tree, characterized by good overall adaptability, growth, development, and strong vitality characteristics [9,10].

*K. paniculata* has been the subject of some phytochemical studies concerning various plant extracts, such as ethanol [2,11,12,13,14,15], methanol [1,13,14,15], benzene/ethanol [13,14,15], and formaldehyde [13], as solvents establishing the different groups of natural components. Phenolic derivatives based on gallic acid, catechins, and phenolic acids are isolated and determined from ethanol (vacuum) extracts of fresh leaf parts [11]. Flavonoids, phenolic acids, and sterols were examined from the ethyl acetate fraction of the *K. paniculata* flowers [16]. Primary (fatty acids, carbohydrates) and secondary metabolites (methyl gallate, ethyl gallate, flavonoids, and their glycosides, sterols, and saponins) are established from 70% ethanol extract of air-dried powdered aerial parts [2,12].

In a few studies, GC-MS analysis was performed in order to identify the chemical composition of the extracts [1,13,14,15]. The authors have analyzed several extracts of bark, wood, branches, leaves, and roots using GC-MS analysis; however, it was not implemented for the identification of the chemical composition of the plant flowers. They found different groups of active components (fatty acids, phenols, mono and triterpenes, sterols, vitamin E, and others). In a previous study by our team, four essential oils from the aerial parts of *K. paniculata* have been isolated by hydrodistillation, for the first time, and they were identified by GC-MS analysis [17]. We found a rich content of different classes of compounds, as follows: aliphatic oxygenated compounds, oxygenated sesquiterpenes, sesquiterpene hydrocarbons, aliphatic hydrocarbons, and diterpenes as predominant groups.

The biological activities exhibited by plants are due to their composition, certain groups of compounds, and the possible interactions between them. The fractions of methanol leaf extract in *K. paniculata* were distinguished by antibacterial and antifungal activity against *Staphylococcus aureus*, *Bacillus subtilis*, and *Pyricularia grisea* [1]. The ethanol extracts obtained from the *K. paniculata* aerial parts showed antibacterial (against *E. coli*) and antimalarial (against chloroquine-sensitive and chloroquine-resistant *Plasmodium falciparum*) activities. Some compounds (such as methyl gallate and ethyl gallate) were among the biologically active secondary metabolites of the plant [2].

Galenic preparations of the *K. paniculata* leaf extracts exhibited high efficiency against six pathogenic microorganisms causing several diseases in humans, such as the following: *Enterococcus faecalis*, *Proteus mirabilis*, *Seracia marcescens*, *Salmonella typhimurium*, *Campylobacter jejuni*, and *Escherichia coli* [5]. 

The antitumor activity of *K. paniculata* extracts has been poorly studied. Kumar et al. [18] reported on the DNA protective effect of the methanol leaf extract and its hexane fraction. The antineoplastic activity of the carotenoid fraction of *K. paniculata* flowers was determined by Zhelev et al. [19]. The authors established low cytotoxicity against human hepatocarcinoma cell lines (HepG2) and human breast cancer cells (MDA-MB-231).

Considering the literature research, it is clear that the species may be a source of valuable biologically active compounds. Still, at the same time, it was evident that in this regard, there were gaps regarding their action and application. In addition, phytochemical research on this species in Bulgaria is scarce, which led us to expand our knowledge on the plant extracts from different parts of this tree species. The present study investigated the qualitative and quantitative composition of the ethanol extracts of the aerial parts of *K. paniculata* and evaluated their in vitro antitumor and antibacterial activities.

## 2. Results and Discussion

### 2.1. Chemical Compounds of Dry Ethanol Extracts from Aerial Parts of K. paniculata

The obtained dry (under vacuum) ethanol extracts’ yields were 0.6521 (2.6084) g for stem bark, 0.624 (2.4960) g for leaves, and 0.516 (2.0640) g for flowers. The extracts were viscous liquids with a dark brown color and characteristic ointment. The three samples were analyzed via GC-MS analysis. The chemical compounds (with their peaks) are presented in Table 1 and Figure 1.

Forty components were identified in the flower extract, representing 98.70% of the total content. Eighteen of them were in a concentration above 1%. The main components (over 3%) were as follows: pyrogallol (20.86%), *α*-terpinyl acetate (16.42%), *α*-terpinyl isobutanoate (10.32%), ethyl decanoate (5.85%), phenyl ethyl butanoate (3.89%), *γ*-terpineol (3.78%), *α*-selinene (3.36%), and *β*-selinene (3.01%). Qu et al. [16] first identified, in *K. paniculata* flowers, nine components in the ethyl acetate fraction by column chromatography and spectral analysis. The components are the following: sitosterol glucoside, gallic acid, kaempferol, luteolin, kaempferol-3-O-(6″-acetyl)-*β*-d-glucopyranoside, hyperoside-2″-O-acetyl, hyperoside-2″-O-galloyl, hyperoside, and kaempferol-3-O-d-glucopyranoside. In our previous study related to the essential oil composition of *K. paniculata* flowers, there were 38 phytochemicals identified, with twelve main compounds. The five common components for the two types of extracts have been proven, namely the following: *β*-caryophyllene, lauric acid, palmitic acid, oleic acid, and tetracosane [17].

In the leaf extract, 50 components were identified, which represents 97.83% of the total content. Twenty-two of them were found in a concentration above 1% and the major ones (over 3%) were the following ten: *α*-terpinyl acetate (20.24%), phenyl ethyl hexanoate (9.05%), *α*-terpinyl isobutanoate (4.77%), linoleic acid (4.32%), *β*-caryophyllene (3.86%), (3Z)-hexenyl 2-methyl butanoate (3.78%), (2E,4E)-nonadienol (3,53%), lavandulol acetate (3.40%), phenyl ethyl 2-methylbutanoate (3.21%), and epi-*β*-bisabolol (3.03%). In the analyzed fractions of methanol extract from the dry leaves of *K. paniculata,* Ghahari et al. [1] found a smaller number (between two and seven) of the major components (over 3%), of which only linoleic acid (4.69%) was among those found by us. Wang et al. [14] isolated 13 active substances in both ethanol and methanol leaf extracts and 32 in benzene/ethanol leaf extract in which the best represented is ethyl gallate—a phenolic compound with antitumor activity. Andonova et al. [17] identified 49 components in the leaf essential oil from golden rain trees, of which six of them are the major ones (above 3%), different from those in the ethanol leaf extract. Only palmitic acid (2.89%), lauric acid (<1%), and *β*-caryophyllene (<1%) are common.

Fifty components were identified in the bark extract, representing 98.63% of the total content, and twenty-nine of them were in concentrations above 1%. The main components (over 3%) were as follows: neryl acetate (12.37%), (3Z)-hexenyl 2-methyl butanoate (8.15%), (2E, 4E)-nonadienol (4.68%), phenyl ethyl 2-methylbutanoate (3.84%), and *α*-terpinyl acetate (3.56%). Yang et al. [13] analyzed the ethanol bark extract of *K. paniculata* using GC-MS analysis and identified the components palmitic acid, linoleic acid, and ethyl oleate. The first two were also present in our extract in similar concentrations. The same authors reported data on the components of other types of bark extracts (formaldehyde, phenyl alcohol, and benzene alcohol extracts), where the concentrations of palmitic acid methyl ester (1.21%), vitamin E (0.31%), sorbitol (0.25%), and dihydrojasmone (0.18%) were identified. The authors found that the main components of the bark extracts were oleic acid (38.72%), lauric acid (5.90%), and acetic acid (4.41%), which were identified by TD-GC-MS analysis. GC-MS analysis of the bark essential oil of the golden rain tree conducted by Andonova et al. [17] identified thirty-six components with nine major ones. Six compounds of all identified were comparable with those in the ethanol extract-palmitic acid (3.20%), *β*-caryophyllene (1.81%), phytol (1.80%), oleic acid (1.03%), lauric acid (<1%), and tetracosane (<1%). 

A comparative analysis of the chemical composition of the studied extracts showed that the one obtained from the flowers was dominated by pyrogallol. It is an odorless substituent and does not form the odor of the extract but instead determines its biological properties, mainly the antioxidant and antimicrobial potential [20]. The extracts of the flowers were also dominated by the monoterpene alcohol γ-terpineol, as well as its esters with acetic and isobutyric acid, which forms the smell of the extract as fresh bergamot-lavender-like (terpinyl acetate), floral (terpinyl isobutyrate), and pine with floral notes (γ-terpineol). According to our findings, the amount of these compounds was lower in the extracts obtained by the plant leaves and bark.

The differences in the identified components in our study, compared with those reported in the literature, are due to the plant’s growing conditions, the technological parameters of the extraction, and the specificity of the used methodology.

The distribution of the components by chemical groups is presented in Figure 2. Oxygenated monoterpene (OM) derivatives predominated in all three of the extracts (flowers 32.49 ± 0.30%, leaves 35.21 ± 0.30% and stem bark 29.84 ± 0.25%), followed by aliphatic oxygen (AO) derivatives and phenylpropanoids (PP). The other groups were less represented, and their distribution can be seen in the figure, as the deviation of the values ranged from 0.07 to 0.1 for sesquiterpene hydrocarbons, from 0.15 to 0.20 for oxygenated aliphatics, 0.01 for aliphatic hydrocarbons, 0.08 to 0.09 for oxygenated sesquiterpenes, and from 0.20 to 0.26 for phenylpropanoids.

In our previous study, the distribution of the components in the different parts of *K. paniculata* showed some differences as aliphatic and oxygenated hydrocarbons, and sesquiterpenes represented the main part of the isolated essential oils [17].

The distribution of the functional groups concerning the total percentage content of *K. paniculata* ethanol extracts is presented in Figure 3. The group of esters had the highest percentage in all three of the studied extracts, followed by alcohols. An exception was the high content of phenols in the flowers (21.13 ± 0.20%), compared to the other two plant parts, which were present in a low percentage. The groups of acids, phenols, ketones, lactones, and aldehydes were very poorly represented in the examined extracts, as shown in Figure 3. The arrangement of the compounds by the functional groups was related to the manifested biological activities of the extracts. Gabrielli et al. [21] pointed out that phenols, followed by alcohols, aldehydes, ketones, ethers, and hydrocarbons, were of primary importance for the activity of the essential oils.

### 2.2. Antitumor Activity of the K. paniculata Ethanol Dry Extracts

The antiproliferative activity of the *K. paniculata* ethanol extracts obtained from the different plant parts was examined on two tumor cell lines—HT-29 and PC3. The two cell lines were not randomly selected. The human colon adenocarcinoma HT-29 cell line is widely used to study the biology of human colon cancers and showed many characteristics of mature intestinal cells [22]. Another cell line, PC3, is also valuable in carcinogenesis. Prostate cancer is the primary malignancy in men and the second leading cause of cancer-related deaths [23]. The obtained results are shown in Figure 4 and Table 2. Improved antiproliferative activity of the flower extract over the other two extracts (IC_50_-21.44 µg/mL) on the cell line HT-29 was observed. Less pronounced activity (over two times) on the other cell line PC3 (IC_50_-58.76 µg/mL) was also demonstrated. The leaf extract showed almost the same activity as the flower extract on the HT-29 cell line (IC_50_-23.63 µg/mL), while prostate cancer cells were less sensitive to this extract (IC_50_–80.56 µg/mL). The bark extract showed weak inhibition effects on the cell lines (IC50–339.4 µg/mL and 182.8 µg/mL for HT-29 and PC3 cell lines, respectively). As can be seen from the graphs (Figure 4E,F), the antiproliferative activity of the total bark extract was dose-dependent for both cell lines. It strongly resembled the antiproliferative effect of cisplatin, the antitumor standard in the present study. The total leaf and flower extracts affected cell growth only at low concentrations, and had almost the same values at higher concentrations at over 60 mg/mL for the HT-29 (Figure 4A,C) and over 125 mg/mL for PC3 (Figure 4B,D). As a possible reason for this, we can point out the differences in the chemical composition of the plant parts and the ethanol extracts obtained from them, especially the presence of the high content of pyrogallol in the composition of flowers. Pyrogallol is compared to antibiotics and also has antioxidant properties [20]. For example, Ahn et al. [24] reported the antitumor mechanisms of pyrogallol that showed significant cytotoxicity and reduced the number of colonies in Hep3B and Huh7 cells. Other authors revealed that phenols determined the antitumor effect of plant extracts on various tumor cell lines (including HT-29) [25,26]. 

Compared to the findings in our study, Zhelev et al. [19], using the MTT-test, found that the carotenoid fraction from *K. paniculata* flowers demonstrate relatively low cytotoxicity to HepG2 (human hepatocarcinoma) and MDA-MB-231 (human breast cancer cells), such as the HepG2 cell line, is more sensitive. The research, in this case, was related to the cytotoxicity of carotenoids and did not investigate their antiproliferative activity. Several articles have examined the ability of different *K.*
*paniculata* extracts to protect various DNA structures from damaging factors. In the study by Kumar et al. [18,27], the methanol extracts and different fractions from the leaves showed a DNA protective effect in Calf thymus/pUC18, as authors associated its activity with the polyphenol constituents within it. In addition, Kumar and Kaur [28] established the potential of those extracts to inhibit lipid peroxidation and 4-nitroquinoline-1-oxide (4NQO)-induced genotoxicity. In vitro cytotoxicity assay on another *Koelreuteria* species (*K. elegans)* showed the promising anticancer activity of two phenols (from butanol fraction), methyl gallate and austrobailignan, against MCF-7 cell lines, which also reduced the cell proliferation of it [29].

### 2.3. Antimicrobial Activity of the K. paniculata Ethanol Dry Extracts

The results for the tested amounts of the extracts (100 μL, 150 μL) on nine pathogenic strains of microorganisms are presented in Table 3 and Figure 5. The bark extract was the most effective against the Gram-positive bacteria *Bacillus subtilis* ATCC 6633 (18 mm inhibition zone, IZ), *Bacillus cereus* NCTC 10,320 (14 mm IZ), and against the Gram-negative bacteria *Pseudomonas aeruginosa* ATCC 6027 (14 mm IZ) and *Proteus vulgaris* ATCC 6380 (8 mm IZ) at the higher tested concentration of the extract. The inhibitory zone of *K. paniculata* flower extract was quite similar against *P. vulgaris* (10 mm IZ), *B. subtilis* (14 mm IZ), and *B. cereus* (14 mm IZ). On the other hand, the *K. paniculata* leaf extract did not inhibit the test cultures against the Gram-negative bacterium *E. coli* ATCC 8739. 

The differences in IZ values could be explained by the content of pyrogallol and terpineol esters. It is known that the activity on the main components of aromatic products (essential oils, extracts) was arranged in the following sequence: phenols > alcohols > aldehydes > ketones > esthers > hydrocarbons [30].

There was limited information concerning the antimicrobial properties of *K. paniculata*, as the reports were mainly about extract obtained from the plant’s leaves. This is the first paper studying the antibacterial activity of extracts obtained from *K. paniculata* flowers and stem barks. Ghahari et al. [1] reported the antibacterial activity of *K. paniculata* methanol extract from the leaves against *B. subtilis* and *S. aureus*. Zazharskyi et al. [5] investigated the antimicrobial potential (with inhibition zone above 8 mm) of ethanol extracts from golden rain tree extracts against different pathogens, such as the following: *E. faecalis*, *P. mirabilis*, *S. marcescens*, *S. typhimurium*, *C. jejuni*, and *E. coli*; the last of which was the most sensitive microorganism. The authors did not find activity against the tested *P. aeruginosa* compared to the findings reported in our study. Ethyl and methyl gallate were the investigated phenols demonstrated in the study by Mostafa et al. [2]. They were reported as promising antimicrobial (against *E. coli*) and antimalarial (against chloroquine-sensitive plasmodia-*Plasmodium falciparum*) agents.

The antimicrobial activity of different plants is influenced by the chemical composition of the plant and the concentration and conditions of obtaining the extracts. For example, Ham et al. [31] reported that neryl acetate had significantly strong and selective antibacterial activity against Gram-negative fish pathogens. Therefore, the presence of the component in the stem bark extracts could be the reason for its antimicrobial potential. Another study revealed the *α*-terpinyl acetate essential oil and extracts showed high antimicrobial effect against fungi, dermatophytes, bacteria and Candida yeasts [32]. The strain differences between the test cultures may also be relevant to the reported results [33].

## 3. Materials and Methods

### 3.1. Plant Material Collection and Identification

The samples from the aerial parts of *K. paniculata* (stem bark, leaves, and flowers—Figure 6) were collected between May and July 2020 in Plovdiv, Bulgaria (42°8′9.9492″ N, 24°44′31.8048″ E), and botanically identified by Prof. D-r. I. Dimitrova-Dyulgerova (Department of Botany, Faculty of Biology, University of Plovdiv “Paisii Hilendarski”). The voucher specimen (No 060436) has been deposited in the Herbarium of the Agricultural University, Plovdiv, Bulgaria (Herbarium SOA).

### 3.2. Preparation of Dry Plant Extracts

Collected fresh and washed plant materials, after maceration (were ground into fine particles using a home grinder), were subjected to two serial extractions. For the elimination of non-polar compounds, chloroform was used (≥99% extra pure, Karl Roth, Germany) as the first solvent, and ethanol (96%, Ph. Eur., extra pure, Karl Roth, Germany) as a second solvent, to study the active and polar compounds. The extracts were obtained in a ratio of 1:10 (plant material:solvent) to complete exhaustion of the herb for 10 days with intermittent stirring. In this study, 400 g of fresh plant material was soaked in 4L of solvent. The supernatant from the chloroform extract was filtered using Whatman filter paper No. 1 (Sigma-Aldrich, Germany), and the residues were used for a second extraction. To concentrate the extracts, a rotary evaporator was used (Buchi, Rotavapor R-300) at 50 °C. Only the ethanol extracts were used for the present study. The dry extracts were collected in a vial and stored at 4 °C in the dark for further use for GC-MS analysis, antitumor, and antimicrobial tests.

### 3.3. Cell Lines, Test-Microorganisms and Nutrient Media

The PC3 (ATCC^®^ CRL-1435™, human prostate adenocarcinoma) and HT-29 (ATCC^®^ HTB-38™, human colon adenocarcinoma) cell lines were obtained from the American Type Culture Collection (ATCC, Manassas, VA, USA). Dulbecco’s modified Eagle medium (DMEM), fetal bovine serum (FBS), antibiotics (penicillin and streptomycin), and the disposable consumables were supplied by Orange Scientific, Braine-l’Alleud, Belgium.

The following Gram-positive bacteria: *Listeria monocytogenes* NCTC 11994, *Staphylococcus aureus* ATCC 25093, *Bacillus subtilis* ATCC 6633, and *Bacillus cereus* NCTC 10320, and the following Gram-negative bacteria: *Escherichia coli* ATCC 8739, *Salmonella enterica* subsp. *enterica* serovar *abony* NCTC 6017, *Pseudomonas aeruginosa* ATCC 6027, *Proteus vulgaris* ATCC 6380, and *Klebsiella* (clinical isolate) were used in this study. The National Bank supplied the strains for industrial microorganisms and cell cultures. The following selective bacteriological media were used: Listeria Oxford agar base with an additive containing cycloheximide (Biolife); Endo agar (Sigma-Aldrich, Germany); Leifson agar (Merck); Baird-Parker agar base (Biolife) with yolk-tellurite additive and plate mount agar (Merck), Chlorhexidine (Sigma-Aldrich, Germany).

### 3.4. Gas Chromatography-Mass Spectrometry (GC-MS) and GC-FID Analyses

The GC-MS analysis was carried out with an Agilent 7890A gas chromatograph with an HP-5MS capillary column (30 m length, 0.32 mm in diameter, 0.25 µm film-coating thickness) coupled to a mass spectral detector Agilent MSD 5975C with helium as the carrier gas (1.0 mL/min). The temperature regime was in the range from 100 to 300 °C (100 °C, 2 min retention, increase to 180 °C with 15 °C/min, 1 min retention, increase to 300 °C with 5 °C/min, 10 min retention); injector and detector temperatures = 250 °C; mass-detector scan range-m/z = 50–550; injected sample volume-1 μL in flow split ratio 20:1. The compounds were identified by comparing retention times and relative Kovats (RI) indices with those of standard substances and mass spectral data from the Golm Metabolome Database (GMD) [34] and NIST’08 (National Institute of Standards and Technology, USA) (https://www.nist.gov/nist-research-library/reference-format-nist-publications, accessed on 10 February 2021). The experiment was carried out in triplicate.

### 3.5. Antitumor Activity Assay

The antitumor activity testing was performed on cell cultures from two human cell lines using the standard MTT-dye reduction assay, described by Mosmann [35]. The assay is based on the metabolism of the tetrazolium salt MTT to insoluble formazan by mitochondrial reductases. The formazan concentration can be determined spectrophotometrically. The measured absorption is an indicator of the cell viability and metabolic activity. The used cell lines were routinely grown as monolayer in 75 cm^2^ tissue culture flasks in DMEM high-glucose (4.5 g/L), supplemented with 10% FBS and antibiotics. Cultures were maintained at 37.5 °C in a humidified atmosphere under 5% CO_2_. Cells were plated at a density of 1 × 10^3^ cells in 100 µL in each well of the 96-well flat-bottomed microplates and allowed to adhere for 24 h before treatment with the test compounds. A concentration range from 2 to 1000 μg/mL (double increasing manner) was applied for 72 h. The formazan absorption was registered using a microplate reader at λ = 540 nm. Cisplatin (Sigma-Aldrich, Germany) was used as a standard in the assay.

### 3.6. Antimicrobial Activity Assay

The antibacterial activity was determined by modifying the agar diffusion method by measuring the inhibition zones of pathogen growth around metal rings, into which a certain amount of test material was introduced. Selective media for the test cultures were inoculated with pathogen suspensions prepared from a 24-h culture on PCA. From a suitable ten-fold dilution of the suspension, the melted and cooled to 45–50 °C selective media was inoculated. After solidifying the media, sterilized metal rings with a diameter of Ø = 6 mm were placed on their surface, in which 0.10 and 0.15 μL of the extract were imported, respectively. Test cultures were incubated at 37 °C. The diameter (mm) of the growth inhibition zones of the test cultures was measured at 24 and 48 h, and a comparative assessment of their antibacterial activity was made. The final DMSO content was 5% (*v*/*v*), and this solution was used as a negative control. For positive control, chlorhexidine was used (100 μL). The experiments were performed in triplicate [36].

### 3.7. Statistical Analysis

The data of the antimicrobial activity test were analyzed and presented as mean values ± standard deviation (SD). Statistical analysis was carried out using Excel software. A one-way analysis of variance (ANOVA) was performed, and significant differences between samples were determined by applying the Tukey’s honestly significant difference (Tukey “HSD”) test, which is used to test differences among sample means for significance. Tukey “HSD” is considered to be a multiple comparison procedure that is used in order to test the significant differences between all possible pairs of mean values on a variable for groups of research samples. Antitumor activity was expressed as IC_50_ value (concentration required for 50% inhibition of cell growth), calculated using non-linear regression analysis (GraphPad Software, San Diego, CA, USA). The statistical analysis included the application of ANOVA, followed by Bonferroni’s post hoc test. The lowest level of statistical significance was accepted as *p* < 0.05. The measurements in the GS/MS analysis were performed in triplicate and the results were presented as the mean value of the individual measurements with the corresponding standard deviation (SD), using Microsoft Excel.

## 4. Conclusions

In conclusion, the present study demonstrated the antitumor and antimicrobial potential of dry ethanol extracts of *K. paniculata* flowers, leaves, and stem bark. The antitumor activity against two cell lines (HT29-human colon adenocarcinoma and PC3-human prostate adenocarcinoma) and the antimicrobial potential against some pathogenic bacteria (*Pseudomonas aeruginosa* ATCC 6027, *Proteus vulgaris* ATCC 6380 and *Bacillus cereus* NCTC 10320) of *K. paniculata* ethanol extracts were investigated for the first time here. Significant antiproliferative activity was found for the flower and leaf ethanol extracts against the HT-29 cell line. The antibacterial activity (dose-dependent) was determined by the extracts of stem bark and flowers against Gram-positive strains of *Bacillus subtilis* ATCC 6633, and *Bacillus cereus* NCTC 10320, and Gram-negative strains of *P. vulgaris* ATCC 6380, and *P. aeruginosa* ATCC 6027. The leaf ethanol extracts inhibited only *E. coli* ATCC 8739 bacterial growth. Fifty-six components were identified in the studied aerial plant parts, among which the best represented, over 10% were pyrogallol (in the flowers), *α*-terpinyl acetate (in the leaves and flowers), neryl acetate (in the stem bark), and *α*-terpinyl isobutanoate (in the flowers). The oxygenated monoterpenes (by chemical groups) and the esters (functional groups) were the best-represented groups in all of the three extracts. Some of the compounds found in the extracts suggest a possible antioxidant potential. Future research should focus on the radical scavenging ability of extracts, as well as on the mechanism of action of proven antitumor activity. *K. paniculata* could be considered as a potential source of biologically active substances with application in pharmaceutical and food production. They would also be useful for the treatment of cancer, microbial infections, as well as for the production of natural preservatives to extend the shelf-life of food.

## Figures and Tables

**Figure 1 plants-10-02715-f001:**
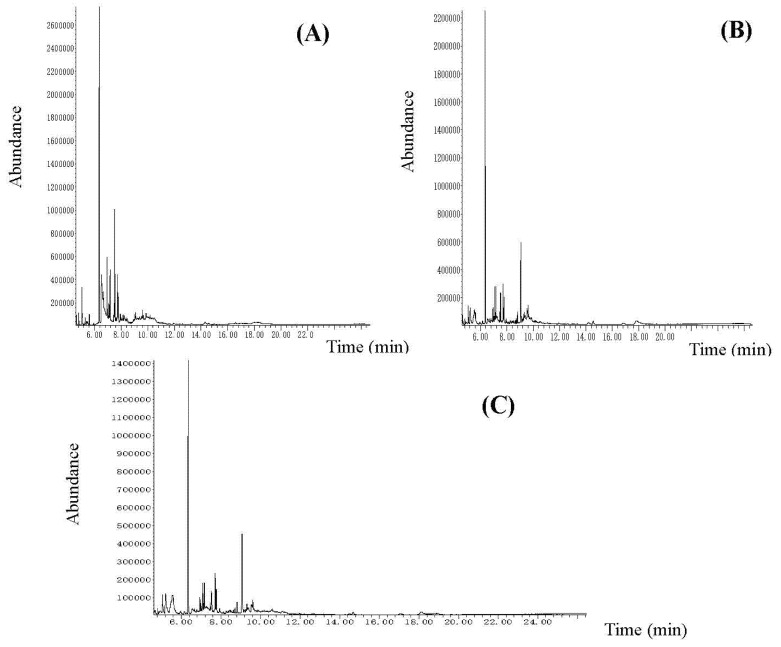
The GC-MS chromatograms of the analyzed dry ethanol extracts from *K. paniculata*: (**A**) flower extract, (**B**) leaf extract, (**C**) stem bark extract.

**Figure 2 plants-10-02715-f002:**
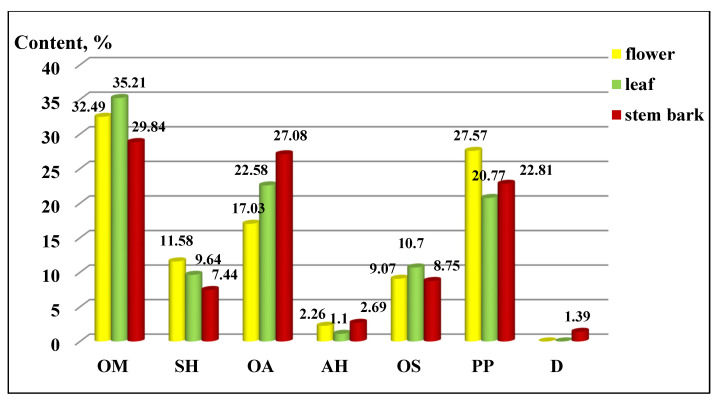
Composition by chemical groups from aerial parts in *Koelreuteria paniculata* ethanol extracts (%): AH—Aliphatic hydrocarbons; OA—Oxygenated aliphatics; OM—Oxygenated monoterpenes; SH—Sesquiterpene hydrocarbons; OS—oxygenated sesquiterpenes; PP—Phenylpropanoids; D—Diterpenes.

**Figure 3 plants-10-02715-f003:**
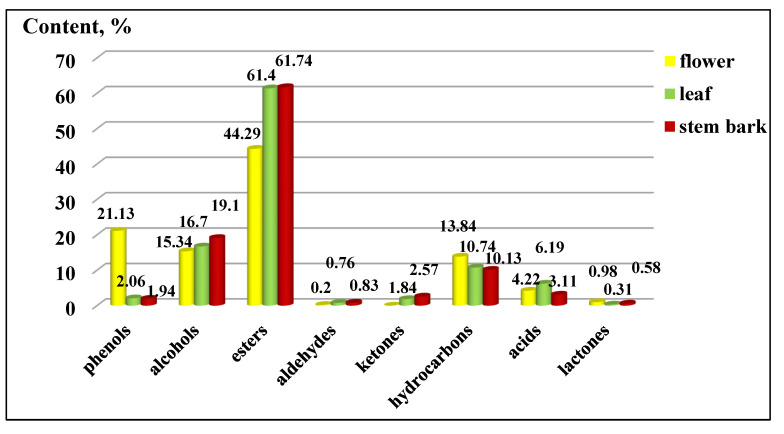
Composition by functional groups from *Koelreuteria paniculata* ethanol extracts (%): phenols; alcohols; esters; aldehydes; ketones; hydrocarbons, acids; lactones. The deviations in the values are in the range of 0.5 to 0.9% statistical error.

**Figure 4 plants-10-02715-f004:**
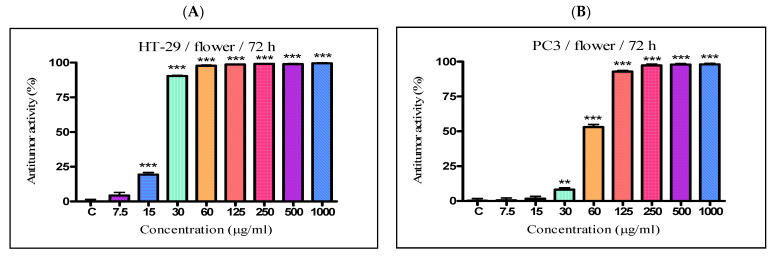
In vitro antiproliferative activity of ethanol extracts from three plant parts of *K. paniculata*. MTT assay was performed after 72 h. (**A**,**C**,**E**) show data for antiproliferative activity on cell line HT-29, of flower, leaf, and bark, respectively. (**B**,**D**,**F**) indicate results obtained on cell line PC3, of the same plant parts respectively. All samples were analyzed in triplicates. Values are represented as mean ± SD; One-way ANOVA followed by post hoc test using Tukey’s multi-group comparison was performed: * *p* < 0.05, ** *p* < 0.01, *** *p* < 0.001.

**Figure 5 plants-10-02715-f005:**
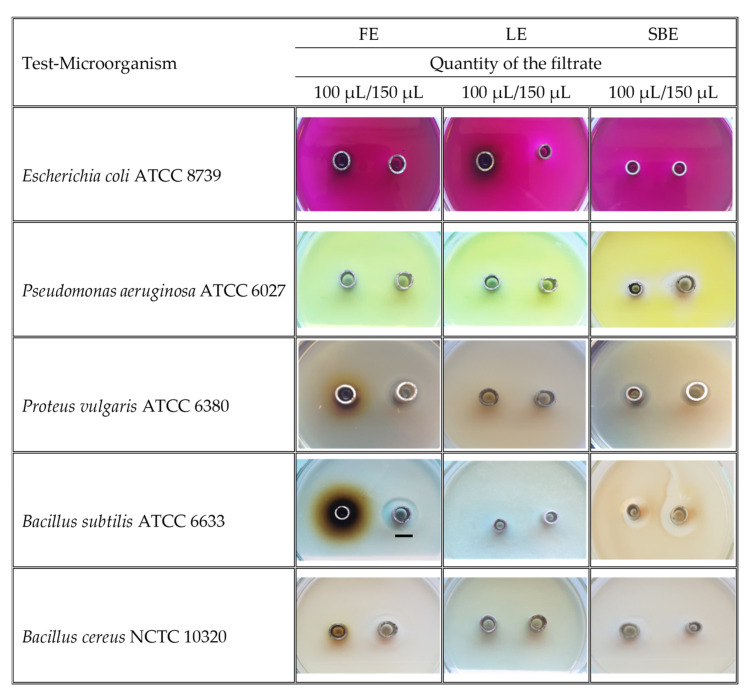
Images of the growth inhibition zones (over 6 mm) of dry ethanol extracts of *Koelreuteria paniculata* aerial parts; FE—Flower extract; LE—Leaf extract; SBE—Stem Bark extract. Scale bar indicated 10 mm.

**Figure 6 plants-10-02715-f006:**
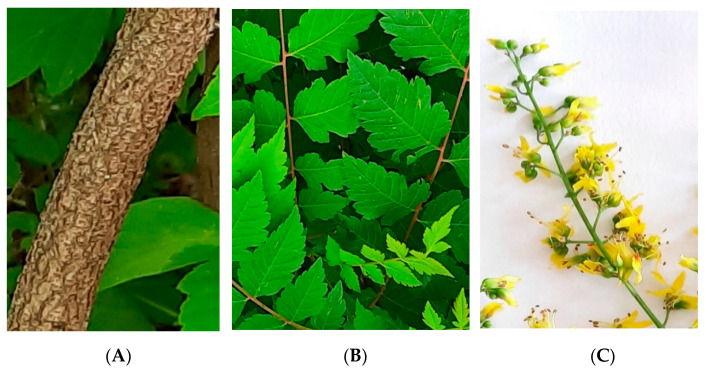
Aerial parts from *Koelreuteria paniculata*: (**A**) stem bark, (**B**) leaves, (**C**) flowers (photos taken by the authors).

**Table 1 plants-10-02715-t001:** Chemical compounds of the ethanol dry extracts of *Koelreuteria paniculata* aerial parts, (mean ± SD).

Peak	RT	RI_calc_	RI_lit_	Compound	*K. paniculata*, % of TIC			Compounds Identification
					Flower	Leaf	Stem Bark	
1	4.49	1174	1170	Furfuryl butanoate	nd	nd	nd	
2	4.68	1190	1186	(3Z)-Hexenyl butanoate	0.81 ± 0.0	0.49 ± 0.0	0.97 ± 0.01	GMD
3	4.79	1197	1199	*γ*-terpineol	3.78 ± 0.10	0.85 ± 0.01	0.55 ± 0.0	GMD
4	5.10	1206	1202	(2E,4E)-Hexadienol isobutanoate	0.13 ± 0.0	2.00 ± 0.07	2.66 ± 0.08	GMD
5	5.20	1218	1217	(2E,4E)-Nonadienol	0.20 ± 0.0	3.53 ± 0.09	4.68 ± 0.11	GMD
6	5.26	1225	1223	Methyl nonanoate	0.58 ± 0.0	0.22 ± 0.0	0.44 ± 0.0	GMD
7	5.30	1233	1228	Nerol	nd	0.49 ± 0.0	1.71 ± 0.05	GMD
8	5.41	1231	1229	(3Z)-Hexenyl 2-methyl butanoate	0.36 ± 0.0	3.78 ± 0.10	8.15 ± 0.20	GMD
9	5.60	1288	1288	Lavandulol acetate	0.47 ± 0.01	3.40 ± 0.09	2.02 ± 0.06	GMD
10	6.11	1316	1315	*δ*-terpinyl acetate	0.12 ± 0.0	0.71 ± 0.0	0.93 ± 0.01	GMD
11	6.23	1345	1346	*α*-terpinyl acetate	16.42 ± 0.51	20.24 ± 0.07	3.56 ± 0.09	GMD
12	6.35	1359	1360	Neryl acetate	0.15 ± 0.0	0.33 ± 0.0	12.37 ± 0.40	GMD
13	6.40	1362	1361	Pyrogallol	20.86 ± 0.70	2.02 ± 0.07	1.91 ± 0.05	NIST’08
14	6.55	1366	1363	*α*-methyl benzyl butyrate	nd	0.30 ± 0.0	0.82 ± 0.0	GMD
15	6.64	1375	1378	(3Z)-Hexenyl hexanoate	nd	0.79 ± 0.0	0.78 ± 0.0	GMD
16	6.82	1386	1386	Isobutyl phenylacetate	nd	1.13 ± 0.04	1.69 ± 0.04	GMD
17	6.94	1394	1395	Ethyl decanoate	5.85 ± 0.12	1.40 ± 0.05	1.47 ± 0.04	GMD
18	7.01	1401	1400	n-tetradecane	2.13 ± 0.07	1.08 ± 0.04	2.65 ± 0.07	GMD
19	7.08	1416	1417	*β*-caryophyllene	2.61 ± 0.07	3.86 ± 0.08	2.91 ± 0.07	GMD
20	7.12	1432	1433	*α*-*trans-*bergamotene	2.45 ± 0.07	0.95 ± 0.01	0.93 ± 0.01	GMD
21	7.16	1439	1438	Phenyl ethyl butanoate	3.89 ± 0.08	2.94 ± 0.08	2.89 ± 0.08	GMD
22	7.26	1453	1453	Geranyl acetone	nd	1.80±	2.53 ± 0.07	GMD
23	7.41	1465	1466	Linalool isovalerate	nd	0.92 ± 0.01	1.42 ± 0.03	GMD
24	7.47	1469	1468	n-dodecanol	2.21 ± 0.07	0.99 ± 0.01	1.90 ± 0.04	GMD
25	7.51	1472	1471	*α*-terpinyl isobutanoate	10.32 ± 0.40	4.77 ± 0.10	3.56 ± 0.09	GMD
26	7.55	1476	1479	*α*-curcumene	nd	0.45 ± 0.0	nd	GMD
27	7.64	1481	1482	*γ*-curcumene	nd	0.58 ± 0.0	nd	GMD
28	7.72	1486	1485	Phenyl ethyl 2-methylbutanoate	0.75 ± 0.0	3.21 ± 0.08	3.84 ± 0.09	GMD
29	7.76	1488	1489	*β*-selinene	3.01 ± 0.07	2.60 ± 0.07	2.63 ± 0.07	GMD
30	7.97	1497	1498	*α*-selinene	3.36 ± 0.08	0.66 ± 0.0	0.72 ± 0.0	GMD
31	8.15	1511	1514	*β*-curcumene	nd	0.33 ± 0.0	0.15 ± 0.0	GMD
32	8.26	1556	1552	Lauric acid	1.93 ± 0.03	0.88 ± 0.0	0.36 ± 0.0	NIST’08
33	8.41	1562	1560	Geranyl butanoate	0.81 ± 0.0	0.55 ± 0.0	0.78 ± 0.0	GMD
34	8.50	1568	1570	Octyl hexanoate	0.40 ± 0.0	0.62 ± 0.0	0.51 ± 0.0	NIST’08
35	8.56	1576	1573	Decyl butyrate	0.64 ± 0.0	0.37 ± 0.0	0.44 ± 0.0	NIST’08
36	8.71	1611	1611	Tetradecanal	0.20 ± 0.0	0.74 ± 0.0	0.82 ± 0.0	GMD
37	8.83	1623	1622	Isobutyl cinnamate	nd	1.22 ± 0.02	1.50 ± 0.02	GMD
38	9.07	1640	1639	Phenyl ethyl hexanoate	1.71 ± 0.03	9.05 ± 0.10	8.67 ± 0.07	GMD
39	9.17	1649	1650	*β*-eudesmol	0.42 ± 0.0	0.90 ± 0.01	0.69 ± 0.0	GMD
40	9.26	1654	1652	*α*-eudesmol	0.27 ± 0.0	0.72 ± 0.0	1.12 ± 0.02	GMD
41	9.37	1662	1661	Dihydro-eudesmol	0.35 ± 0.0	1.35 ± 0.02	1.93 ± 0.03	GMD
42	9.61	1671	1670	Epi*-β-*bisabolol	2.19 ± 0.07	3.03 ± 0.07	1.51 ± 0.02	GMD
43	9.67	1675	1674	*β*-bisabolol	2.88 ± 0.07	1.44 ± 0.02	1.66 ± 0.02	GMD
44	9.86	1690	1691	(Z)-*α*-*trans*-Bergamotol	2.00 ± 0.06	2.52 ± 0.07	1.00 ± 0.02	GMD
45	10.21	1699	1698	(2Z,6Z)-Farnesol	0.84 ± 0.0	0.51 ± 0.0	0.72 ± 0.0	GMD
46	10.50	1706	1704	*δ*-dodecalactone	0.97 ± 0.01	0.30 ± 0.0	0.57 ± 0.0	GMD
47	10.82	1718	1718	Methyl eudesmate	nd	0.45 ± 0.0	1.18 ± 0.02	GMD
48	11.93	1826	1825	(E)-Nerolidyl isobutyrate	nd	0.39 ± 0.0	nd	GMD
49	14.49	1953	1948	Phytol	nd	nd	0.14 ± 0.0	GMD
50	14.61	1971	1966	Palmitic acid	1.85 ± 0.05	0.86 ± 0.0	1.16 ± 0.02	GMD
51	14.73	1992	1990	Ethyl palmitate	0.29 ± 0.0	nd	nd	GMD
52	16.67	2018	2019	(6E,10Z)-Pseudo phytol	nd	nd	0.42 ± 0.0	GMD
53	16.85	2026	2024	Isopropyl hexadecanoate	nd	0.79 ± 0.0	0.25 ± 0.0	GMD
54	16.94	2029	2030	(6E,10E)-Pseudo phytol	nd	nd	0.81 ± 0.0	GMD
55	18.05	2133	2132	Linoleic acid	0.16 ± 0.0	4.32 ± 0.04	1.55 ± 0.02	GMD
56	18.28	2142	2141	Oleic acid	0.23 ± 0.0	nd	nd	GMD
57	21.58	2400	2400	Tetracosane	0.10 ± 0.0	nd	nd	GMD
**Total identified, %**					**98.70**	**97.83**	**98.63**	

RT—Retention time; RI_calc_—Kovats retention index, calculated by authors; RI_lit_—Kovats retention index by literature data; TIC—Total ion current; nd—not detected; NIST’08—National Institute of Standards and Technology, Gaithersburg, MD, USA; GMD—Golm Metabolome Database.

**Table 2 plants-10-02715-t002:** In vitro antiproliferative activity of the ethanol dry extracts of *Koelreuteria paniculata* aerial parts.

	IC_50_ of Mean ± SD (µg/mL)	
	HT-29	PC3
**Flower extract**	21.44 ± 0.20	58.76 ± 0.53
**Leaf extract**	23.63 ± 0.22	80.56 ± 0.75
**Stem Bark extract**	339.4 ± 1.31	182.8 ± 1.65
**Cisplatin**	2.5 ± 0.08	1.01 ± 0.03

IC_50_ determined following 72 h treatment with ethanol extracts. Antitumor activities were expressed as IC_50_ values (extract concentrations (µg/mL) required for 50% inhibition of cell growth), calculated using non-linear regression analysis (GraphPad Software, San Diego, CA, USA). Results were calculated from three measurements and expressed as mean ± SD. Cisplatin was used as standard to confirm the suitability of the used antitumor method.

**Table 3 plants-10-02715-t003:** Zones of growth inhibition (mm) of dry ethanol extracts of *Koelreuteria paniculata* aerial parts.

Test-Microorganism	QF	FE	LE	SBE	CH
**Gram-negative bacteria**					
*Escherichia coli* ATCC 8739	150 μL	- *	9.02 ^a^ ± 0.01	-	28.05 ^b^ ± 0.01
	100 μL	-	6.02 ^a^ ± 0.01	-	
*Salmonella enterica* NCTC 6017	150 μL	-	-	-	24.04 ± 0.03
	100 μL	-	-	-	
*Klebsiella* (clinical isolate)	150 μL	-	-	-	21.03 ± 0.03
	100 μL	-	-	-	
*Pseudomonas aeruginosa* ATCC 6027	150 μL	-	-	14.02 ^c^ ± 0.01	21.03 ^c^ ± 0.03
	100 μL	-	-	5.03 ^c^ ± 0.01	
*Proteus vulgaris* ATCC 6380	150 μL	10.04 ^d^ ± 0.02	-	8.04 ^d^ ± 0.01	-
	100 μL	6.05 ^d^ ± 0.04	-	6.02 ^d^ ± 0.01	
**Gram-positive bacteria**					
*Staphylococcus aureus* ATCC 25093	150 μL	-	-	-	25.03 ± 0.03
	100 μL	-	-	-	
*Bacillus subtilis* ATCC 6633	150 μL	14.03 ^e^ ± 0.03	-	18.04 ^e^ ± 0.02	39.04 ^f^ ± 0.03
	100 μL	11.03 ^e^ ± 0.04	-	11.03 ^e^ ± 0.02	
*Bacillus cereus* NCTC 10320	150 μL	14.0 6 ^g^ ± 0.03	-	14.03 ^g^ ± 0.02	-
	100 μL	-	-	6.02 ^g^ ± 0.01	
*Listeria monocytogenes* NCTC 11994	150 μL	-	-	-	21.06 ± 0.02
	100 μL	-	-	-	

Data are presented as means ± SD (standard deviation). ^a–g^ Means in a row not sharing the same superscript letter are significantly different at *p* < 0.05 (Tukey’s HSD test). QF—Quantity of the filtrate; FE—Flower extract; LE—Leaf extract; SBE—Stem bark extract; CH—Chlorhexidine; *—No inhibitory activity was observed.

## Data Availability

Not applicable.

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
