# Peer review of "Chemical Compounds, Antitumor and Antimicrobial Activities of Dry Ethanol Extracts from Koelreuteria paniculata Laxm"

_plants, 2021, doi:10.3390/plants10122715_

Round 1

Reviewer 1 Report

Dear Authors,

The work titled "Chemicals, Antitumor and Antimicrobial Effects of Dry Ethanol Extracts from Koelreuteria paniculata Laxm." Due to the innovative nature of research on the antitumor activity of plant extracts from Koelreuteria paniculata flowers and leaves and their antiproliferative activity on tumor cell lines and antibacterial activity on pathogenic bacterial strains, it deserves publication. However, the authors did not avoid the flaws and errors that should be eliminated before going to print. Here are the comments: 1) The abstract of the work should contain the aim, scope of the research and the most important conclusion, and not a suggestion for further research. 2) The introduction to the work should summarize the aim of the work and the research hypothesis (against the null hypothesis), which should be verified later in the work. 3) The research methodology should be before discussing the results, and it is only after this chapter, which is illogical. 4) The authors compare the results of their work directly with the results in the literature on the subject. There is not much of this discussion, however, and I propose to separate the discussion as a separate chapter and expand it considerably. 5) not all test results are sufficiently interpreted. The authors refer to the statistical analysis, but it is not practically used or well presented (the bar of the HSD values ​​in the tables). Descriptive statistics should also be better used. Here, the authors only refer to statistical deviations, and yet descriptive statistics also have other indicators such as: kurtosis, skewness, range, coefficient of variation, which belongs to the scattering measures, so it is used to study the degree of variation in the value of the variable. A high value of the coefficient means a large diversification of the trait and indicates the heterogeneity of the studied population, while its low value indicates low variability of the trait and homogeneity of the studied population. It is therefore necessary to make better use of these indicators for a better interpretation of the results obtained and to demonstrate the assumed objectives of the work. 6) There is a bug in table 2, please check it. There are also no indications as to the statistical differences referred to by the authors 7) Figure 4 is illegible and incomprehensible, the reader cannot read anything from it, the markings are invisible. 8) The chemical compositions in all three extracts should be better compared, not only by chemical groups 9) conclusions should be generalizing and summarizing. 10) The work should contain a chapter towards the future 11) the literature should be supplemented with the newest items in order to conduct a good discussion.

Author Response

Dear Reviewer,

Thank you for your helpful comments and suggestions on our manuscript, which helped to make our paper more rigorous and valuable. We have tried our best to revise our manuscript according to the reviewers’’ detailed suggestions. The main corrections are pointed in the revised manuscript and below and as follows:

Dear Authors,

The work titled "Chemicals, Antitumor and Antimicrobial Effects of Dry Ethanol Extracts from Koelreuteria paniculata Laxm." due to the innovative nature of research on the antitumor activity of plant extracts from Koelreuteria paniculata flowers and leaves and their antiproliferative activity on tumor cell lines and antibacterial activity on pathogenic bacterial strains, it deserves publication. However, the authors did not avoid the flaws and errors that should be eliminated before going to print.

Response: We thank Reviewer 1 for her/his appreciation of our manuscript.

1) The abstract of the work should contain the aim, scope of the research and the most important conclusion, and not a suggestion for further research.

Response: The abstract was modified accordingly.

2) The introduction to the work should summarize the aim of the work and the research hypothesis (against the null hypothesis), which should be verified later in the work.

Response: Thank you for the suggestion. Summary of the aim of the work was done in the introduction part.

3) The research methodology should be before discussing the results, and it is only after this chapter, which is illogical.

Response : According to the journal instructions, the results and the discussion parts should be followed by the material and methods.

4) The authors compare the results of their work directly with the results in the literature on the subject. There is not much of this discussion, however, and I propose to separate the discussion as a separate chapter and expand it considerably.

Response : We thank the reviewer for the comments and suggestions. According to the journal instructions, “Results and discussion” could be together or separated. In our case, we decided to present them together but we expanded the discussion related to the obtained results as suggested.

5) not all test results are sufficiently interpreted. The authors refer to the statistical analysis, but it is not practically used or well presented (the bar of the HSD values in the tables). Descriptive statistics should also be better used. Here, the authors only refer to statistical deviations, and yet descriptive statistics also have other indicators such as: kurtosis, skewness, range, coefficient of variation, which belongs to the scattering measures, so it is used to study the degree of variation in the value of the variable. A high value of the coefficient means a large diversification of the trait and indicates the heterogeneity of the studied population, while its low value indicates low variability of the trait and homogeneity of the studied population. It is therefore necessary to make better use of these indicators for a better interpretation of the results obtained and to demonstrate the assumed objectives of the work.

Response : Extra explanation about the Tukey’s honestly significant difference (Tukey “HSD”) test was added.

5) There is a bug in table 2, please check it. There are also no indications as to the statistical differences referred to by the authors

Response: Thank you for your comment. The calculations were repeated and the new standard deviations were added. Cisplatin units were also corrected.

6) Figure 4 is illegible and incomprehensible, the reader cannot read anything from it, the markings are invisible.

Response : Figures 4 was corrected accordingly.

7) The chemical compositions in all three extracts should be better compared, not only by chemical groups

Response: Thank you for the suggestion and consequently the discussion on the chemical composition of the extracts was enlarged in the text.

8) conclusions should be generalizing and summarizing.

Response : Conclusion was modified as suggested by the reviewer.

9) The work should contain a chapter towards the future.

Response : Having in mind the suggestion, a part regarding the future application as added into the text.

10) the literature should be supplemented with the newest items in order to conduct a good discovery.

Response : Thank you for the suggestion. Recent publication were added. As our study is the first one considering that species and due to lack of studies regarding  Koelreuteria paniculata anti-tumor activitiy, there is not way for comparison.

Reviewer 2 Report

The workmanuscript entitled "Chemical compounds, antitumor and antimicrobial activities of dry ethanol extracts from Koelreuteria paniculata" seems interesting to me since it presents the revaluation of extracts of wild plants for therapeutic purposes. However, I think that for this work to be accepted in the magazine "Plants" it should be expanded. The studies are not enough and nothing has been deepened on the mechanism of action of these extracts. Therefore, I ask the authors the following questions:
1. The chemical composition of the different extracts, Figures 2 and 3, was carried out more than once? In the figures do not appreciate the errors.
2. The different compositions found between flower, leaf and stem bark and their relationship with the biological activity in the two cell lines tested (colon and prostate cancer) are not discussed.
3. Cell viability was determined by the MTT method and it is known that this method in plant extracts containing polyphenols gives false positives. Why weren't the SRB or Resazurin methods used?
4. Figure 4 would be better understood if% cell viability were placed on the ordinate axis.
5. With the results presented, it is not known whether these extracts cause the death of cancer cells by apoptosis, necrosis or autophagy. It is also not known whether the effect of plant extracts is selective for cancer cells.
6. The mechanism of action by which this plant acts on cancer cells is not indicated.
7. It has not been determined whether these extracts have antioxidant power.
I think that this work to be accepted must be completed with additional experiments.

Author Response

Dear Reviewer,

Thank you for your helpful comments and suggestions on our manuscript, which helped to make our paper more rigorous and valuable. We have tried our best to revise our manuscript according to the reviewers’’ detailed suggestions. The main corrections are pointed in the revised manuscript and below and as follows:

The workmanuscript entitled "Chemical compounds, antitumor and antimicrobial activities of dry ethanol extracts from Koelreuteria paniculata" seems interesting to me since it presents the revaluation of extracts of wild plants for therapeutic purposes. However, I think that for this work to be accepted in the magazine "Plants" it should be expanded. The studies are not enough and nothing has been deepened on the mechanism of action of these extracts. Therefore, I ask the authors the following questions:

Response: We thank Reviewer 2 for appreciations of our manuscript and for the constructive suggestions. Below are the responses to her/his questions.

1) The chemical composition of the different extracts, Figures 2 and 3, was carried out more than once? In the figures do not appreciate the errors.

Response: The experiment was carried out in triplicate and standard deviations (SD) were added in to the main body of the manuscript.

2) The different compositions found between flower, leaf and stem bark and their relationship with the biological activity in the two cell lines tested (colon and prostate cancer) are not discussed.

Response : Thank you for the suggestions. Discussion about that was added to the manuscript.

3) Cell viability was determined by the MTT method and it is known that this method in plant extracts containing polyphenols gives false positives. Why weren't the SRB or Resazurin methods used?

Response: The method is widely used to determine antitumor activity in plant extracts and is therefore used in the present study. Below we appended several recent manuscript published in high impact factor journals applying MTT method.

Published manuscripts applying MTT method and plnt extracts

Abid, F., Saleem, M., Muller, C. D., Asim, M. H., Arshad, S., Maqbool, T., & Hadi, F. (2020). Anti-Proliferative and Apoptosis-Inducing Activity of Acacia Modesta and Opuntia Monocantha Extracts on HeLa Cells. Asian Pacific journal of cancer prevention : APJCP, 21(10), 3125–3131. https://doi.org/10.31557/APJCP.2020.21.10.3125

Al-Hamwi, Mohammad, Maha Aboul-Ela, Abdalla El-Lakany, and Salam Nasreddine. "Anticancer Activity of Micromeria fruticosa and Teucrium polium Growing in Lebanon." Pharmacognosy Journal 13, no. 1 (2021). DOI:10.5530/pj.2021.13.15

Ashraf, Kamran, Hasseri Halim, Siong Meng Lim, Kalavathy Ramasamy, and Sadia Sultan. "In vitro antioxidant, antimicrobial and antiproliferative studies of four different extracts of Orthosiphon stamineus, Gynura procumbens and Ficus deltoidea." Saudi journal of biological sciences 27, no. 1 (2020): 417-432. https://doi.org/10.1016/j.sjbs.2019.11.003.

Kowalik, K.; Paduch, R.; Strawa, J.W.; Wiater, A.; Wlizło, K.; Waśko, A.; Wertel, I.; Pawłowska, A.; Tomczykowa, M.; Tomczyk, M. Potentilla alba Extracts Affect the Viability and Proliferation of Non-Cancerous and Cancerous Colon Human Epithelial Cells. Molecules 2020, 25, 3080. https://doi.org/10.3390/molecules25133080

Miceli, N.; Cavò, E.; Ragusa, M.; Cacciola, F.; Mondello, L.; Dugo, L.; Acquaviva, R.; Malfa, G.A.; Marino, A.; D’Arrigo, M.; Taviano, M.F. Brassica incana Ten. (Brassicaceae): Phenolic Constituents, Antioxidant and Cytotoxic Properties of the Leaf and Flowering Top Extracts. Molecules 2020, 25, 1461. https://doi.org/10.3390/molecules25061461

Mohammed, H.A.; Al-Omar, M.S.; Khan, R.A.; Mohammed, S.A.A.; Qureshi, K.A.; Abbas, M.M.; Al Rugaie, O.; Abd-Elmoniem, E.; Ahmad, A.M.; Kandil, Y.I. Chemical Profile, Antioxidant, Antimicrobial, and Anticancer Activities of the Water-Ethanol Extract of Pulicaria undulata Growing in the Oasis of Central Saudi Arabian Desert. Plants 2021, 10, 1811. https://doi.org/10.3390/plants10091811

Simas Frauches, N.; Montenegro, J.; Amaral, T.; Abreu, J.P.; Laiber, G.; Junior, J.; Borguini, R.; Santiago, M.; Pacheco, S.; Nakajima, V.M.; Godoy, R.; Teodoro, A.J. Antiproliferative Activity on Human Colon Adenocarcinoma Cells and In Vitro Antioxidant Effect of Anthocyanin-Rich Extracts from Peels of Species of the Myrtaceae Family. Molecules 2021, 26, 564. https://doi.org/10.3390/molecules26030564

Somaida, A.; Tariq, I.; Ambreen, G.; Abdelsalam, A.M.; Ayoub, A.M.; Wojcik, M.; Dzoyem, J.P.; Bakowsky, U. Potent Cytotoxicity of Four Cameroonian Plant Extracts on Different Cancer Cell Lines. Pharmaceuticals 2020, 13, 357. https://doi.org/10.3390/ph13110357

Tesfaye, S.; Braun, H.; Asres, K.; Engidawork, E.; Belete, A.; Muhammad, I.; Schulze, C.; Schultze, N.; Guenther, S.; Bednarski, P.J. Ethiopian Medicinal Plants Traditionally Used for the Treatment of Cancer; Part 3: Selective Cytotoxic Activity of 22 Plants against Human Cancer Cell Lines. Molecules 2021, 26, 3658. https://doi.org/10.3390/molecules26123658

Venkatachalapathy, D.; Shivamallu, C.; Prasad, S.K.; Thangaraj Saradha, G.; Rudrapathy, P.; Amachawadi, R.G.; Patil, S.S.; Syed, A.; Elgorban, A.M.; Bahkali, A.H.; Kollur, S.P.; Basalingappa, K.M. Assessment of Chemopreventive Potential of the Plant Extracts against Liver Cancer Using HepG2 Cell Line. Molecules 2021, 26, 4593. https://doi.org/10.3390/molecules26154593

4) Figure 4 would be better understood if % cell viability were placed on the ordinate axis.

Response: Thank you for the suggestion, but the present way in our study is also representative.

5) With the results presented, it is not known whether these extracts cause the death of cancer cells by apoptosis, necrosis or autophagy. It is also not known whether the effect of plant extracts is selective for cancer cells.

Response: Thank you for the constructive remark that will improve the quality of present work. In this case, we investigate the antiproliferative activity of plant extracts on 2 cell lines, or their ability to inhibit reproduction of the cells. As this is an initial (first) study on the antitumor activity of the studied plant species, we are in the process of conducting additional experiments with other extracts and on other tumor cell lines. We also plan to study the mechanisms of action (apoptosis, necrosis, autophagy, etc.)

6) The mechanism of action by which this plant acts on cancer cells is not indicated.

Response: Thank you for pointing that. This is the first study describing the anti-tumor activity of Koelreuteria paniculata. We are working further on the mechanism of acting.

7) It has not been determined whether these extracts have antioxidant power.
I think that this work to be accepted must be completed with additional experiments.

Response: Thank you very much for your comment; it is a very good point. We consider continuing with the experiments related to the plant species, so the antioxidant activity is currently being investigated and reported as the future study.

Round 2

Reviewer 1 Report

Dear Authors,

For Authors,

Dear Authors,

The authors considered all my comments as a Reviewer. They supplemented the abstract and the methodology of work, improved the quality of the drawings, completed the tables with the results, supplemented the description of the results and interpreted them better. They also separated the discussion as a separate chapter and extended it with new aspects. They supplemented the application and also supplemented the references. Thus, they significantly increased the quality of work.

Author Response

Dear Authors,

The authors considered all my comments as a Reviewer. They supplemented the abstract and the methodology of work, improved the quality of the drawings, completed the tables with the results, supplemented the description of the results and interpreted them better. They also separated the discussion as a separate chapter and extended it with new aspects. They supplemented the application and also supplemented the references. Thus, they significantly increased the quality of work.

Response: We are grateful to Reviewer 1 for the positive comments and we would like to thank him/her for suggestions that made the manuscript more rigorous and attractive.

Reviewer 2 Report

I still think the work is incomplete. To be accepted, it is essential that they carry out additional experiments in which it is determined whether these extracts have antioxidant power. Likewise, it is necessary if the antiproliferative power is due to programmed death or necrosis.
In relation to the viability studies with the MTT assays, it has been proven that at high concentrations of extracts, it gives false positives.

The chemical composition table of the extracts does not show any errors or deviations in the data.

Author Response

Reviewer 2:

I still think the work is incomplete.

To be accepted, it is essential that they carry out additional experiments in which it is determined whether these extracts have antioxidant power.

Likewise, it is necessary if the antiproliferative power is due to programmed death or necrosis.

In relation to the viability studies with the MTT assays, it has been proven that at high concentrations of extracts, it gives false positives.

The chemical composition table of the extracts does not show any errors or deviations in the data.

Response:

We thank the Reviewer for her/his comments.

Here are the responses to the comments made by the reviewer:

-             Standard deviations have been added to the table showing the chemical composition of the extracts;

-             Regarding the MTT test: Indeed, there is evidence of a false positive result at high concentrations of extracts from some plant species. These false-positive results are due to a reduction in MTT tetrazolium salt (3-(4,5-dimethylthiazol-2-yl)-2,5-diphenyltetrazolium bromide) to insoluble purple formazan crystals by reducing agents included in the extract and not by mitochondrial reductases of living cells. In our case, however, the antitumor activity is inversely proportional to the measured optical density, which in the area of the highest concentrations is significantly lower than the negative control, in which cell viability is assumed to be 100% or 0% - antitumor effect. This is proof that in the extracts studied by us in the concentration range from 1000 to 7.5 µg / ml no false positive result is observed.

-             We have also planned studies of the antitumor potential of extracts with other methods, such as Flow cytometry analysis of apoptosis and cell cycle to determine the mechanism by which they act, such as cytotoxins or antiproliferative agents.

-             Thank you for the recommendation for research on antioxidant activity again. Such experiments are ongoing and initial results were obtained for radical-scavenging activity of extracts by DPPH- and ABTS- аssays. They demonstrate really high antioxidant potential. Work continues on two other methods for antioxidant activity, as well as on the content of flavonoids and phenolic acids, which will be the subject of our next publication.